# Nexus-Gen: Unified Image Understanding, Generation, and Editing via Prefilled Autoregression in Shared Embedding Space

## Abstract

Unified multimodal generative models aim to integrate image understanding and generation abilities, offering significant advantages in harnessing multimodal corpora, particularly interleaved text-image data. However, existing unified models exhibit limitations in image synthesis quality, autoregressive error accumulation, and image editing capability. In this work, we propose Nexus-Gen, a novel architecture that unifies image understanding, generation, and editing tasks in a shared image embedding space. This shared space serves as a bridge for the autoregressive and diffusion models, which seamlessly integrates their complementary strengths in cross-modal modeling. To mitigate the severe error accumulation during autoregressive embedding prediction, we propose a novel prefilled autoregression strategy that aligns training-inference dynamics by prefilling input sequences with learnable embeddings. After multi-stage and multi-task training on our constructed large-scale dataset with 26.3 million samples, Nexus-Gen achieves state-of-the-art performance on the evaluation benchmarks spanning image understanding, generation and editing tasks. All models, datasets, and codes will be released to facilitate further advancements across the field.

## 1 Introduction

Multimodal generative modeling has emerged as a pivotal frontier in AI research Bai et al. (2025). Multimodal large language models (MLLMs) demonstrate notable competence in image understanding, while diffusion models lead in image generation. To further harness the potential of vision-language modeling, particularly the synergistic benefits across modalities, recent studies focused on MLLMs with unified modality architectures Chen et al. (2025b); Tong et al. (2024b). The core advantage of unified modeling lies in the efficient data utilization and cross-modal representation learning, enabling the integration of multiple cross-modal capabilities. Such integration proves effective for complex tasks like image editing, visual reasoning, and reasoning-enhanced image generation, positioning unified MLLMs as a crucial pathway toward unified general intelligence.

Integrating image understanding and generation within a unified framework while enabling joint optimization remains a fundamental challenge for unified MLLMs. Prior works, including Chameleon Team (2024), Janus-Pro Chen et al. (2025b), and Emu3 Wang et al. (2024), predominantly adopt autoregressive models coupled with variational autoencoders (VAEs) Kingma et al. (2013). These frameworks employ autoregressive models to predict visual embeddings, which are fed into VQ-VAE Sun et al. (2024a) or VAE decoders for image generation. However, they underperform state-of-the-art diffusion models Podell et al. (2024); Esser et al. (2024); Labs (2024) in image synthesis. This gap is attributed to the lack of pixel-level image modeling capabilities of autoregressive models. Another category of methods, such as SEED-X Ge et al. (2024) and MetaMerph Tong et al. (2024b), predict image embeddings autoregressively and employ diffusion models for image generation. A key limitation of these works is the unresolved error accumulation phenomenon during autoregressive token-by-token generation of continuous embeddings. Additionally, the extensibility of these frameworks to downstream tasks (e.g., image editing) remains insufficiently validated.

To leverage the rich pretrained knowledge of autoregressive LLMs and diffusion models, this paper proposes Nexus-Gen, a framework that unifies image understanding, generation, and editing tasks

within a shared continuous image embedding space. As illustrated in Figure 1(a), this embedding space serves as a pivotal interface between the autoregressive model and diffusion vision decoder. It also plays a vital role in preserving information integrity while maintaining embedding versatility across diverse tasks. For image understanding task, input images are encoded into the unified space to predict textual outputs. For image generation and editing tasks, the autoregressive model generates the target image embeddings within the space, which are decoded into images by vision decoders. Crucially, we uncover that autoregressive prediction of image embeddings suffers from severe error accumulation. To address this, we propose the prefilled autoregression strategy, which prefills the input sequence with learnable embeddings to align training and inference dynamics.

To perform joint optimization across multiple tasks and fully utilize the multimodal corpus, we construct a large-scale dataset of 26.3 million samples and propose a multi-stage training strategy for Nexus-Gen. The first stage conducts multi-task pretraining of the autoregressive model across image understanding, generation, and editing tasks. This process develops unified any-to-any modal prediction capabilities. The second stage adapts the generation vision decoder by fine-tuning it on a high-quality image generation dataset, substituting the textual conditioning with our unified image embeddings. The third stage adapts the editing decoder by fine-tuning it on our curated high-quality ImagePulse dataset to enable dual-stream image embedding inputs. Through multi-stage training, Nexus-Gen achieves superior performance on image understanding, generation and editing tasks. Specifically, it attains scores of 45.7 on the MMMU understanding benchmark Yue et al. (2024) and 0.81 on GenEval generation benchmark Ghosh et al. (2023). Our contributions are as follows:

- We propose Nexus-Gen, a unified model that leverages a unified image embedding space to bridge the capabilities of LLMs and diffusion models.

- We propose a prefilled autoregression strategy that effectively avoids error accumulation during the prediction process of autoregressive models, thereby extending the generative capabilities of autoregressive models.

- We demonstrate the capabilities of Nexus-Gen through extensive experiments. Multiple benchmarks show that Nexus-Gen achieves state-of-the-art performance in image understanding, generation, and editing tasks.

- We curate and release a dataset comprising 26.3 million samples for unified image understanding, generation, and editing tasks to promote research advances.

## 2 RELATED WORKS

Recent advances in unified architectures for image understanding and generation have stimulated research efforts, leading to two dominant paradigms: autoregressive model with VAE and autoregressive model with diffusion models.

**Autoregressive Model with VAE** These methodologies exclusively employ lightweight VAEs Kingma et al. (2013) or VQ-VAEs Sun et al. (2024a) for image decoding, positioning the autoregressive model to modeling image information within the pixel space. Notably, Chameleon Team (2024), Show-O Xie et al. (2024), and Emu3 Wang et al. (2024) utilize a VQ-Tokenizer as the vision decoder, training the LLM on interleaved text-image data for unified modeling. Janus Wu et al. (2025a) and Janus-Pro Chen et al. (2025b) further refine this architecture by decoupling understanding and generation tasks within different encoding spaces. They employ SigLIP Zhai et al. (2023) for understanding-oriented encoding and VQ-Tokenizer for generative encoding, respectively. Crucially, all aforementioned methods rely on autoregressive prediction of subsequent visual tokens. Conversely, Transfusion Zhou et al. (2025), and Janus Flow Ma et al. (2025) adopt diffusion loss to optimize visual token generation.

**Autoregressive Model with Diffusion** Leveraging pre-trained diffusion models as an additional component to synthesize image, these approaches typically yield superior image quality compared to autoregressive model with VAE techniques. The closed-source GPT-4o OpenAI (2025) model exemplifies this architectural paradigm by adopting the workflow: token → [transformer] → [diffusion] → pixels. Representative open-source frameworks, including SEED-X Ge et al. (2024), Emu2 Sun et al. (2024b), and MetaMerph Tong et al. (2024b), adopt SDXL Podell et al. (2024) as the vision

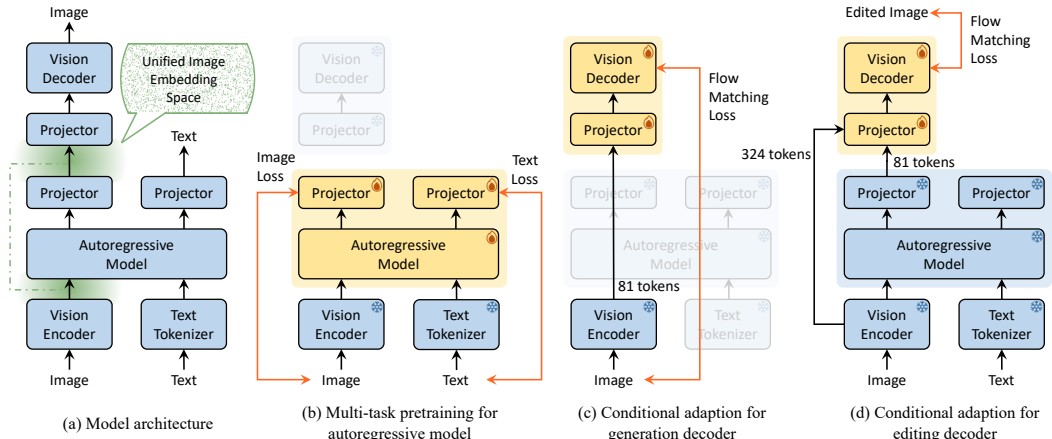

Figure 1: The architecture and the multi-stage training recipe for Nexus-Gen.

decoder while utilizing regression loss to optimize the language LLM's visual prediction capability. Regarding architectural variations under this paradigm, LM-Fusion Shi et al. (2024) and MetaQuery Zhou et al. (2025) both maintain the LLM in a frozen state. The former trains the vision decoder via shared attention mechanisms, while the latter employs learnable queries as an intermediary bridge between the LLM and the diffusion model. In contrast to the aforementioned methods, this work employs a unified embedding space to jointly model image understanding, generation and editing tasks, and designs different conditioning architectures for generation and editing. This design enables the LLM to capture cross-task correlations, facilitating subsequent research into interleaved tasks and reasoning-based multimodal understanding and generation. Additionally, we propose the prefilled autoregression strategy to optimize the generation of continuous image embeddings.

## 3 APPROACH

### 3.1 ARCHITECTURE

The architecture of Nexus-Gen incorporates three core components, which are depicted in Figure 1(a). The vision encoder and decoder are responsible for unified image embedding, while the autoregressive model facilitates unified multimodal context-aware reasoning.

**Unified Image Embedding Space**   Existing research Gu et al. (2025) on multimodal representations has demonstrated that unified embedding training across multiple downstream tasks facilitates a more comprehensive understanding of content information than single-task training. **Thus, we adopt the embedding space of the vision encoder as a unified image embedding space to jointly model image-related tasks.** Images in image understanding, generation and editing tasks are all projected to this embedding space. For image understanding, images are encoded into embeddings, which are then further interpreted by the autoregressive model. For image generation, the autoregressive model generates target image embeddings based on textual descriptions, and the embeddings are subsequently decoded into images by the vision decoder. By integrating both understanding and generation capabilities, our framework enables image editing ability through modifications to the image embeddings. As unified models evolve towards reasoning-intensive and multi-turn conversational paradigms, our unified embedding space allows for the direct reuse of model-generated embeddings in subsequent reasoning or multi-turn conversations.

**Vision Encoder**   We adopt the vision transformer of Qwen2.5-VL-7B-Instruct Bai et al. (2025) as our vision encoder and utilize its vision embedding space as our unified image embedding space. This space is implicitly aligned with textual representation space due to the multimodal abilities of the base model, which facilitates the establishment of robust text-image mapping with minimal training data. Leveraging the dynamic resolution capability of the vision encoder, we can modulate the number of image embedding tokens ($N_E$) by adjusting the input resolution ($H \times W$). This

relationship is formally expressed as:

$$N_E = \left\lfloor \frac{H}{P} \right\rfloor \times \left\lfloor \frac{W}{P} \right\rfloor \tag{1}$$

where $P = 14$ denotes the size of each patch. Higher resolutions produce more token embeddings, with each token corresponding to a smaller spatial region and capturing more low-level features. Conversely, lower resolutions yield fewer tokens where each represents a larger spatial region, resulting in embeddings that encode more high-level features.

**Autoregressive Model**    The autoregressive model is utilized for unified multimodal reasoning, as illustrated in Figure 1(a). Textual inputs are tokenized by the text tokenizer and projected into text embeddings, while visual inputs are encoded into image embeddings through the vision encoder. These text and image embeddings are jointly fed into the autoregressive model to predict the embeddings of output tokens. For image synthesis tasks, the output image embeddings predicted by the model are mapped to the unified image embedding space via a vision projector. For text generation, the output text embeddings are converted into logits by the text projector. In Nexus-Gen, the trainable parameters of the autoregressive model and text projector are initialized from Qwen2.5-VL-7B-Instruct Bai et al. (2025) to inherit the pretrained linguistic abilities. The vision projector is a randomly initialized linear layer for embedding alignment.

To prevent information loss, input images for understanding and editing tasks are encoded to embeddings at their native resolution without downsampling. For output images in generation and editing tasks, there exists a trade-off. Employing more image tokens helps capture finer image details. However, an excessive token count makes the task substantially more difficult for the autoregressive model, leading to degraded generation performance. Through experimental validation, we opt for a token count of 81, ensuring the reconstruction quality of the vision decoder without imposing excessive generation pressure on the autoregressive model.

**Vision Decoder**    To achieve high-fidelity image decoding from model-generated embeddings, we adopt the diffusion transformer of FLUX.1-Dev Labs (2024) as our vision decoder by replacing its native T5 text embeddings Raffel et al. (2020) with our designed conditioning mechanisms. Given the divergent objectives of image generation and editing tasks, we implement specialized conditioning schemes and architectural configurations for the respective decoders.

For image generation, which emphasizes semantic consistency with textual description, the decoder is designed to reconstruct images that are semantically consistent with the 81-token image embeddings produced by the autoregressive model, as presented in Figure 1(c). To condition the diffusion transformer on these embeddings, a two-layer MLP projector is utilized for embedding alignment.

For image editing tasks requiring faithful execution of editing instructions while preserving details in unaltered regions, we propose an editing decoder with dual conditioning mechanisms, as detailed in Figures 1(d). The first condition incorporates 81-token embeddings from the autoregressive model, conveying semantic information of the target image. The second condition integrates 324-token embeddings derived from direct encoding of the input image, preserving fine-grained details of the original content. To effectively model the hierarchical relationships between these two conditions, we introduce a joint attention layer as the embedding projector for the diffusion transformer.

## 3.2 PREFILLED AUTOREGRESSION

When optimizing the autoregressive model, we observe significant error accumulation in the continuous embedding space, attributed to the discrepancy between the training and inference behaviors of autoregressive models. As depicted in Figure 2(a), the model leverages ground-truth preceding tokens for each prediction during teacher-forcing training, whereas at inference time, it relies on autoregressively generated tokens. When processing continuous image embeddings, the model directly predicts biased embeddings and feeds them back as input. This recursive injection of prediction errors propagates and amplifies across subsequent tokens, resulting in suboptimal performance.

Prior research Li et al. (2024) has stated that image token prediction is permutation-invariant, and each image embedding can be derived solely from the caption and positional encoding without depending on preceding embeddings. Thus, we propose the prefilled autoregression strategy to

Figure 2: (a) The naive autoregressive approach exhibits inconsistent behaviors between training and testing phases, leading to error accumulation during inference. (b) We propose a strategy that prefills special image tokens during training and testing, which unifies the computational behaviors across both phases and eliminates error accumulation.

mitigate error accumulation, as shown in Figure 2(b). During training, we initialize all image tokens with $N_E$ learnable embeddings with positional encoding (where $N_E$ is the token quantity). During inference, upon predicting the BOI (beginning of image) token, we prefill the input sequence with the learned embeddings. This enforces alignment between training and inference by preventing error-affected predictions from being recycled into the input, thereby eliminating error accumulation.

## 3.3 DATASET CURATION

To enable Nexus-Gen with unified visual capabilities, we construct a dataset with 26.3 million samples covering image understanding, generation, and editing tasks. The majority of our dataset derives from publicly accessible open-source repositories Zhang et al. (2025); Tong et al. (2024a); Zhao et al. (2024a), which is relabeled to improve annotation quality. However, given the image quality limitations (e.g., aesthetic, artifacts) in existing editing datasets, we additionally construct a high-quality image editing dataset, ImagePulse. We will release all training data after data security and legality checks. The complete dataset construction pipeline is provided in the Appendix.

## 3.4 TRAINING OBJECTIVES

The training of Nexus-Gen encompasses both its autoregressive model and vision decoder components. The autoregressive model undergoes unified multi-task training, generating outputs comprising both text and image embeddings. The loss function for the text and image embeddings is formulated as follows, where $\lambda_1 = 3$, $\lambda_2 = 1.5$, $\lambda_3 = 1.5$ are hyperparameters that control the loss weights.

$$L = L_{\text{Text}} + L_{\text{Image}} = \lambda_1 \cdot L_{\text{CE}} + (\lambda_2 \cdot L_{\text{MSE}} + \lambda_3 \cdot L_{\text{COS}}) \tag{2}$$

For the text tokens, we employ the standard cross-entropy loss for classification, which is defined in Eq. 3. Here, $N_T$ denotes text tokens numbers, $|V|$ represents the vocabulary size, $y_t^{(c)}$ is the ground-truth one-hot encoded token, and $\hat{y}_t^{(c)}$ indicates the predicted probability distribution.

$$L_{\text{CE}} = -\frac{1}{N_T} \sum_{t=1}^{N_T} \sum_{c=1}^{|V|} y_t^{(c)} \log(\hat{y}_t^{(c)}) \tag{3}$$

For the image embeddings, we utilize a composite loss function combining mean squared error and cosine similarity loss Radford et al. (2021). This combination ensures the preservation of detail fidelity while simultaneously enforcing semantic coherence. The loss functions are defined in Eq. 4 and 5, where $N_E$ is image embeddings numbers, $D$ signifies the embedding dimensionality, $\hat{\mathbf{e}}_i$ and $\mathbf{e}_i$ denote the predicted and ground-truth embeddings, respectively.

$$L_{\text{MSE}} = \frac{1}{N_E \cdot D} \sum_{i=1}^{N_E} \|\mathbf{e}_i - \hat{\mathbf{e}}_i\|_2^2 \tag{4}$$

$$L_{\text{COS}} = -\frac{1}{N_E} \sum_{i=1}^{N_E} \frac{\mathbf{e}_i \cdot \hat{\mathbf{e}}_i}{\|\mathbf{e}_i\|_2 \cdot \|\hat{\mathbf{e}}_i\|_2} \tag{5}$$

The vision decoders for generation and editing tasks are trained separately. They both adopt the MSE loss function of flow matching. Given the diffusion transformer $V_\theta$, target image $X_1$ in latent space, noise $X_0 \sim N(0, 1)$, conditions $C$ and timestep $t \sim \mu(0, 1)$, the loss function is defined as:

$$L_{\text{Flow}} = \mathbb{E}\left[ \left\| V_\theta\left(X_t, C, t\right) - \left(X_1 - X_0\right) \right\|^2 \right] \tag{6}$$

### 3.5 TRAINING STRATEGY

We adopt a multi-stage training strategy for Nexus-Gen. The first stage consists of the unified multi-task pretraining and aesthetic fine-tuning for autoregressive model, as is shown in Figure 1(b). The second and third stages conduct conditional adaptation for the generation decoder and editing decoder, which is illustrated in Figure 1(c) and (d). All hyperparameters are listed in the Appendix.

**Multi-Task Pretraining for Autoregressive Model**  The first stage executes unified pretraining and aesthetic fine-tuning of the autoregressive model. Pretraining utilizes the complete dataset of 26.3 million samples, primarily preserving the autoregressive model's inherent text prediction capacity while establishing a visual embedding prediction functionality. Subsequent aesthetic fine-tuning process employs 4.3 million high-quality samples to optimize Nexus-Gen's visual output quality. This high-quality dataset comprises: (1) premium image generation samples Zhang et al. (2025); gogoduan (2025); jackyhate (2024); Chen et al. (2025a), (2) our novel ImagePulse dataset augmented with 0.5 million randomly selected instances from other editing corpora Zhao et al. (2024a); Hui et al. (2024), and (3) 1 million image understanding data Tong et al. (2024a).

**Conditional Adaption for Generation Decoder**  The second stage fine-tunes the generation decoder to harmonize its conditional inputs with the unified image embedding space via an image reconstruction objective. To preserve visual fidelity, this module is exclusively trained on two million high-quality image generation samples Zhang et al. (2025); gogoduan (2025); Chen et al. (2025a).

**Conditional Adaption for Editing Decoder**  The third stage fine-tunes the editing decoder using our ImagePulse dataset, as illustrated in Figure 1(d). This decoder processes dual heterogeneous inputs: 324-token fine-grained embeddings extracted from the input image and 81-token semantic embeddings generated by the autoregressive model.

## 4 EXPERIMENTS

### 4.1 MAIN RESULTS

In this section, we conduct a quantitative evaluation of Nexus-Gen across multiple benchmarks for image understanding, generation, and editing tasks.

**Image Understanding**  For image understanding tasks, we evaluate the model performance on several multimodal understanding benchmarks: MME Fu et al. (2024), SEEDBench Li et al. (2023), MMMU Yue et al. (2024), TextVQA Singh et al. (2019), VQAv2 Goyal et al. (2017), and RealWorldQA XAI (2024). For baseline comparison, we incorporate the following unified models: Seed-X Ge et al. (2024), Chameleon Team (2024), Emu3 Wang et al. (2024), Metamorph Tong et al. (2024b), Show-O Tong et al. (2024b), VILA-U Wu et al. (2024), Janus Wu et al. (2025a), Orthus Kou et al. (2024), Harmon Wu et al. (2025b), Tar Han et al. (2025), Janus-Pro Chen et al. (2025b), LMFusion Shi et al. (2024), and TokenFlow Qu et al. (2025). As is shown in Table 1, the 7B-parameter Nexus-Gen achieves state-of-the-art performance across all evaluated benchmarks. It outperforms other unified models by significant margins, with particular advantages on MME-Cognition and TextVQA. We further evaluate our MLLM baseline Qwen2.5-VL-Instruct 7B. Empirical results demonstrate that Nexus-Gen endows the base autoregressive model with image generation and editing capabilities without incurring significant loss in image understanding performance, while introducing no additional parameters.

Table 1: Evaluation on image understanding benchmarks. Best and second-best results are in bold and underlined, respectively. † The Qwen2.5-VL-Instruct 7B is evaluated as the reference baseline.

| Model | MME-P ↑ | MME-C ↑ | SEED ↑ | MMMU ↑ | TextVQA ↑ | VQAv2 ↑ | RWQA ↑ |
|---|---|---|---|---|---|---|---|
| Qwen2.5-VL 7B † | 1689.0 | 640.3 | 77.4 | 50.6 | 77.9 | 82.3 | 66.0 |
| Seed-X 17B | 1457.0 | - | 66.5 | 35.6 | - | 63.4 | - |
| Chameleon 34B | 604.5 | - | - | 38.8 | - | 69.6 | 39.2 |
| EMU3 8B | 1243.8 | 266.1 | 68.2 | 31.6 | 64.7 | 75.1 | 57.4 |
| MetaMorph 8B | - | - | 71.4 | 41.8 | 60.5 | - | 58.3 |
| Show-O 1.3B | 1097.2 | - | - | 27.4 | - | 69.4 | - |
| VILA-U 7B | 1336.2 | - | 56.3 | 32.2 | 48.3 | 75.3 | 46.6 |
| Janus 1.5B | 1338.0 | - | 63.7 | 30.5 | - | - | - |
| Orthus 7B | 1265.8 | - | - | 28.2 | - | 63.2 | - |
| Harmon 1.5B | 1155.0 | 321.0 | 67.1 | 38.9 | - | - | - |
| Tar 1.5B | 1390.0 | 342.0 | 70.4 | 36.0 | - | - | - |
| Janus-Pro 7B | 1567.1 | - | 72.1 | 41.0 | - | - | 60.0 |
| LMFusion 16B | 1603.7 | 367.8 | 72.1 | 41.7 | - | - | 60.0 |
| TokenFlow-XL 14B | 1551.1 | 371.1 | 72.6 | 43.2 | 62.3 | 77.6 | 56.6 |
| Nexus-Gen 7B (Ours) | 1602.3 | 637.5 | 77.1 | 45.7 | 75.5 | 79.3 | 63.7 |

Table 2: Evaluation of image generation ability on GenEval benchmark. We highlight the best result in bold, and underline the second-best result. Nexus-Gen underwent joint optimization across all three tasks. We subsequently perform instruction tuning on the Blip3o-60k dataset targeting image generation task, resulting in the optimized Nexus-Gen* variant.

| Method | Single ↑ | Two ↑ | Counting ↑ | Colors ↑ | Position ↑ | Color Attr. ↑ | Overall↑ |
|---|---|---|---|---|---|---|---|
| Emu3-Gen 8B | 0.99 | 0.81 | 0.42 | 0.80 | 0.49 | 0.45 | 0.66 |
| SEED-X 17B | 0.97 | 0.58 | 0.26 | 0.80 | 0.19 | 0.14 | 0.49 |
| Show-O 1.3B | 0.95 | 0.52 | 0.49 | 0.82 | 0.11 | 0.28 | 0.53 |
| TokenFlow 14B | 0.95 | 0.60 | 0.41 | 0.81 | 0.16 | 0.24 | 0.55 |
| Orthus 1.5B | - | - | - | - | - | - | 0.58 |
| Janus 1.5B | 0.97 | 0.68 | 0.3 | 0.84 | 0.46 | 0.42 | 0.61 |
| Transfusion 7B | - | - | - | - | - | - | 0.63 |
| Harmon 1.5B | 0.99 | 0.86 | 0.66 | 0.85 | 0.74 | 0.48 | 0.76 |
| Tar 1.5B | - | 0.91 | 0.76 | - | - | 0.51 | 0.76 |
| MetaQuery 7B | - | - | - | - | - | - | 0.80 |
| Janus-Pro 7B | 0.99 | 0.89 | 0.59 | 0.90 | 0.79 | 0.66 | 0.80 |
| Nexus-Gen 7B (Ours) | 0.99 | 0.86 | 0.53 | 0.85 | 0.78 | 0.59 | 0.77 |
| Nexus-Gen* 7B (Ours) | 0.97 | 0.93 | 0.64 | 0.88 | 0.83 | 0.62 | 0.81 |

**Image Generation**  For text-to-image generation tasks, we adopt GenEval Ghosh et al. (2023) as the evaluation benchmark, with assessment metrics covering object fidelity, quantity accuracy, color correctness, attribute matching, and spatial relationships. We compare Nexus-Gen and its instruction-tuned variant Nexus-Gen* (fine-tuned on Blip3o-60k dataset Chen et al. (2025a)) against unified models including Emu3 Wang et al. (2024), SEED-X Ge et al. (2024), Show-O Xie et al. (2024), TokenFlow Qu et al. (2025), Orthus Kou et al. (2024), Janus Wu et al. (2025a), Transfusion Zhou et al. (2025), Harmon Wu et al. (2025b), Tar Han et al. (2025), MetaQuery Pan et al. (2025), and Janus-Pro Chen et al. (2025b). Results in Table 2 reveal that the multi-task jointly-trained Nexus-Gen achieves an overall score of 0.77. After generative-task-specific instruction tuning on Blip3o-60K Chen et al. (2025a) dataset, Nexus-Gen* attains state-of-the-art performance with a score of 0.81. This enhanced model demonstrates advantages in Two Object and Position metrics.

**Image Editing**  For image editing evaluation, we utilize the test set with 1,000 randomly sampled cases from ImagePulse dataset (non-overlapping with training data). On this benchmark, we compare Nexus-Gen against state-of-the-art models: InstructPix2Pix Brooks et al. (2023), MagicBrush Zhang et al. (2023), AnyEdit Yu et al. (2025), UltraEdit Zhao et al. (2024a), OmniGen Xiao et al. (2025), and Step1X-Edit Liu et al. (2025). We employ three complementary metric categories: (1) CLIP-T measures the CLIP image-text similarity between edited image and target caption. (2) L1 measures the pixel-level absolute difference between the edited and ground-truth images. (3) CLIP-O and DINO-O measure the cosine similarity between the edited and ground-truth images using

Table 3: Evaluation of image editing ability on ImagePulse benchmark. We highlight the best results in bold, and underline the second-best result.

| Method | CLIP-T ↑ | L1 ↓ | CLIP-O ↑ | DINO-O ↑ |
|---|---|---|---|---|
| InstructPix2Pix | 0.299 | 0.171 | 0.832 | 0.706 |
| MagicBrush | 0.309 | 0.146 | 0.863 | 0.750 |
| AnyEdit | 0.305 | 0.141 | 0.863 | 0.756 |
| UltraEdit | 0.306 | 0.157 | 0.841 | 0.737 |
| OmniGen | 0.317 | 0.154 | 0.874 | 0.764 |
| Step1X-Edit | 0.317 | 0.142 | 0.879 | 0.779 |
| Nexus-Gen | **0.324** | **0.134** | **0.909** | **0.834** |

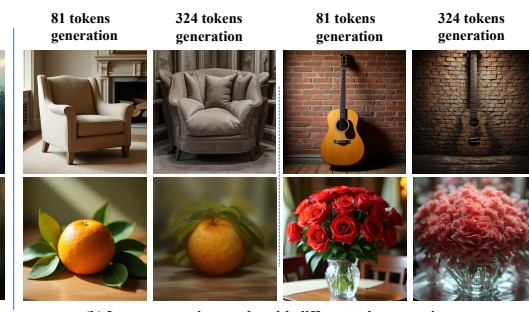

(a) Image reconstruction results with different token quantity.  (b) Image generation results with different token quantity.

Figure 3: Trade-offs in Token Quantity. (a) Image reconstruction results of our generation decoder using 25, 81 and 324 image token embedding. (b) Image generation results from Nexus-Gen trained with 81 and 324 image token embeddings.

their CLIP Radford et al. (2021) and DINO Caron et al. (2021) embeddings. As displayed in Table 3, Nexus-Gen demonstrates superiority across all metrics. This performance advantage confirms Nexus-Gen's enhanced capability to faithfully execute editing instructions and generate outputs with notably improved alignment to both target captions and ground-truth images.

## 4.2 ABLATION STUDIES

**Trade-offs in Token Quantity**    The token quantity for image embeddings exhibits a positive correlation with resolution. Higher resolution produces more tokens, allowing embeddings to retain finer visual details. We consider three standard resolutions: $128 \times 128$, $256 \times 256$ and $512 \times 512$, corresponding to 25, 81 and 324 tokens, respectively. We train the generation decoder at these token quantities for image reconstruction, with results shown in Figure 3(a). The reconstructions with 81 and 324 tokens preserve better global layouts and high-level semantics of source images. Notably, the 324-token approach demonstrates significantly superior detail consistency. In contrast, the 25-token reconstruction exhibits structural distortions and semantic loss relative to the source image.

We further train the autoregressive model at token counts of 81 and 324 and validate image generation quality using above vision decoders. Experimental results in Figure 3(b) demonstrate that the model trained with 81 tokens effectively generates images aligned with textual semantics. However, the 324-token model exhibits severe semantic repetition in generated images, which exhibited compromised quality. This indicates that autoregressive models struggle to accurately predict such extensive image tokens. Consequently, we select token quantity of 81 as the optimal count for both the autoregressive model and generation decoder.

**The Necessity of Editing Decoder**    Although the generation decoder can be directly applied to image editing tasks, we propose the editing decoder to enhance the detail preservation capability in non-edited regions, as displayed in Figure 1 (d). Figure 4 compares the editing performance of both decoders. Given identical inputs, both solutions successfully execute edit instructions. However, the generation decoder fails to maintain details in non-edited areas due to its 81-token reconstruc-

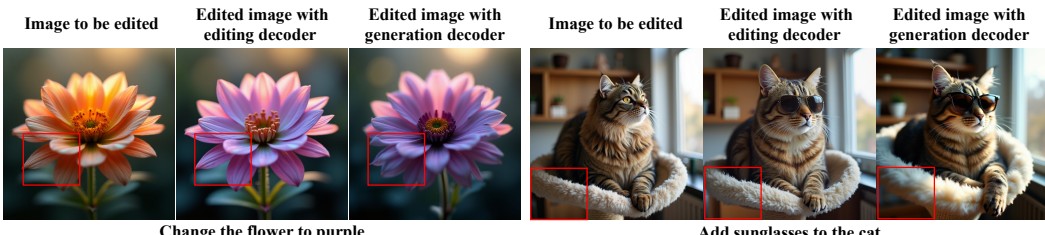

Figure 4: Image editing results of Nexus-Gen with editing and generation decoder.

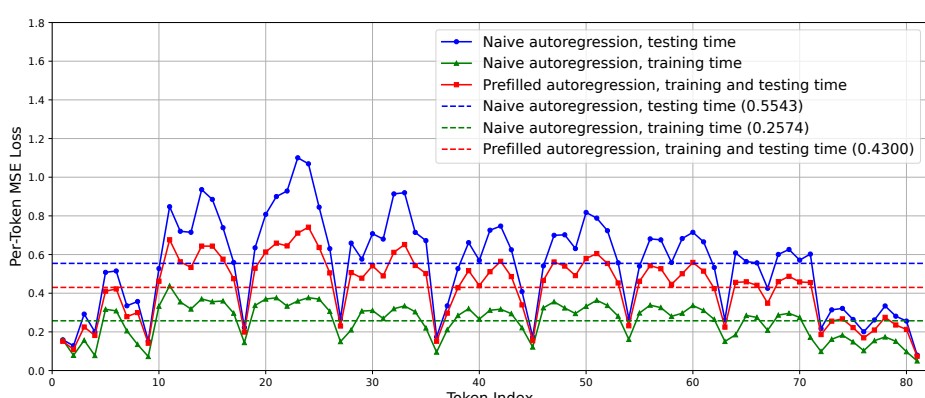

Figure 5: MSE loss comparison between the naive and prefilled autoregression strategy.

tion constraints. In contrast, the editing decoder synergizes the superior 324-token reconstruction capability with the efficient 81-token embedding prediction, achieving higher editing fidelity.

**The Effectiveness of Prefilled Autoregression**    To mitigate the error accumulation issue in naive autoregression paradigm, we propose the prefilled autoregression strategy. Figure 5 compares the MSE losses of image tokens predicted by both approaches. During training, naive autoregression achieves the lowest loss of 0.2574 due to access to preceding ground-truth embedding. During inference, the autoregressive nature causes progressive error accumulation, elevating the average loss to 0.5543. In contrast, our prefilled autoregression strategy maintains consistent training-inference behavior, achieving a significantly lower inference loss of 0.43.

## 5    CONCLUSION AND FUTURE WORKS

In this work, we present Nexus-Gen, a unified model for image understanding, generation, and editing tasks. The core innovation of Nexus-Gen lies in bridging the language reasoning capabilities of LLMs with the image synthesis power of diffusion models through a unified continuous image embedding space. Furthermore, we identify the error accumulation phenomenon during the autoregressive prediction of continuous embeddings and propose prefilled autoregression strategy to mitigate it. To perform joint optimization across multiple tasks, we curate a large-scale dataset of 26.3 million samples and train Nexus-Gen using a multi-stage strategy, which includes the multi-task pretraining of the autoregressive model and conditional adaptations of the generation and editing decoders. Extensive experiments validate Nexus-Gen's state-of-the-art performance across all tasks. While Nexus-Gen successfully unifies the three image tasks, certain limitations warrant attention. The model exhibits compromised robustness to prompt variations during image generation, and more significantly, its capacity for sophisticated visual reasoning remains unexplored. To address these constraints, we will focus on developing Nexus-Gen's advanced applications in complex tasks such as in-context learning and step-by-step vision-language reasoning.

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

# A DATASET CONSTRUCTION DETAILS

## A.1 DATASET DISTRIBUTION

We construct a unified dataset covering image understanding, generation and editing tasks with 26.3 million samples. Detailed dataset distribution is presented in Figure 6.

**Image Understanding** This task is structured with multimodal inputs (image-text pairs) and text-only outputs, which serves as a direct indicator of model's chat and understanding ability. While MLLMs inherently possess such cross-modal reasoning capabilities, this task is still critical during training to prevent capacity degradation. We adopt Cambrian-7M Tong et al. (2024a) as the data source, a comprehensive dataset spanning multiple domains including optical character recognition, general visual question answering, language, counting, code, math and science tasks. To improve data quality, we re-annotate the answers for all samples with Qwen2.5-VL-72B Bai et al. (2025).

**Image Generation** The input for this task is the textual description, and the output is an image. Our data sources comprise Journey DB Pan et al. (2023), AnyWord Tuo et al. (2023), Laion-High-Resolution LAION (2024), EliGen TrainSet Zhang et al. (2025), FLUX-Aes gogoduan (2025), FLUX-T2I jackyhate (2024) and Blip3o-60K Chen et al. (2025a). To enhance annotation diversity, we employ a dual-captioning paradigm via Qwen2.5-VL-72B, generating both concise captions and elaborate descriptions for each image. During training, we stochastically sample these annotations with stratified ratios (20% concise vs. 80% elaborate) to balance brevity and contextual granularity.

**Image Editing** The input for editing task consists of an image and its corresponding editing instruction, and the output denotes the edited image. Our data sources encompass datasets such as HQ-Edit Hui et al. (2024), UltraEdit Zhao et al. (2024a), OmniEdit Wei et al. (2024), and StyleBooth Han et al. (2024). These datasets cover an extensive range of image editing operations, including object-level manipulations, color adjustments, and style transformations. However, these datasets exhibit notable limitations in aesthetic quality. However, they exhibit notable limitations in aesthetic quality and visual fidelity. To address this, we construct the high-quality ImagePulse dataset and integrate it into our training corpus.

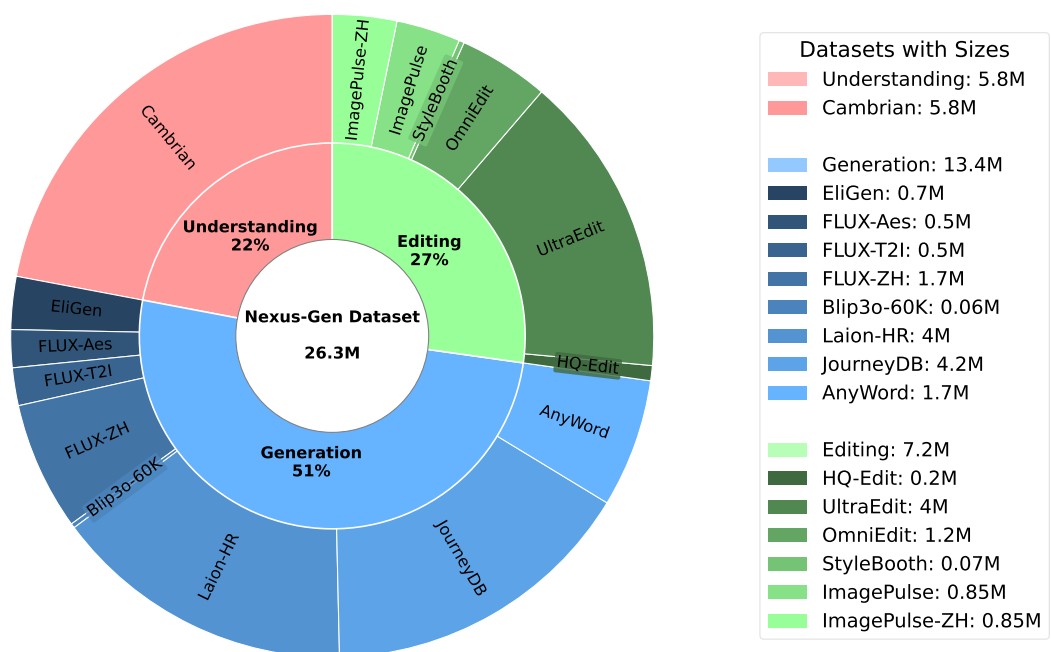

Figure 6: Dataset distribution of our Nexus-Gen dataset.

Table 4: Detailed hyperparameters for training Nexus-Gen. Data ratio refers to the ratio of image understanding data, generation data and editing data.

| Training Phase | Multi-task Pretraining for Autoregressive Model | | Conditional Adaption for Vision Decoder | |
|---|---|---|---|---|
| Training Target | Large Scale Pretraining | Aesthetic Fine-Tuning | Generation Decoder | Editing Decoder |
| Learning Rate | 1e-5 | 1e-5 | 1e-5 | 1e-5 |
| LR Scheduler | Cosine | Cosine | Constant | Constant |
| Batch Size | 512 | 512 | 128 | 128 |
| Total Steps | 100 K | 10 K | 20 K | 10 K |
| Warm-up Steps | 7500 | 800 | 100 | 100 |
| Total Samples (Million) | 26 | 4 | 2 | 1 |
| Data Ratio (Und:Gen:Edit) | 1:2:1 | 1:2:1 | 0:1:0 | 0:0:1 |
| GPU Hours | 43 K | 12 K | 1.3 K | 0.8 K |

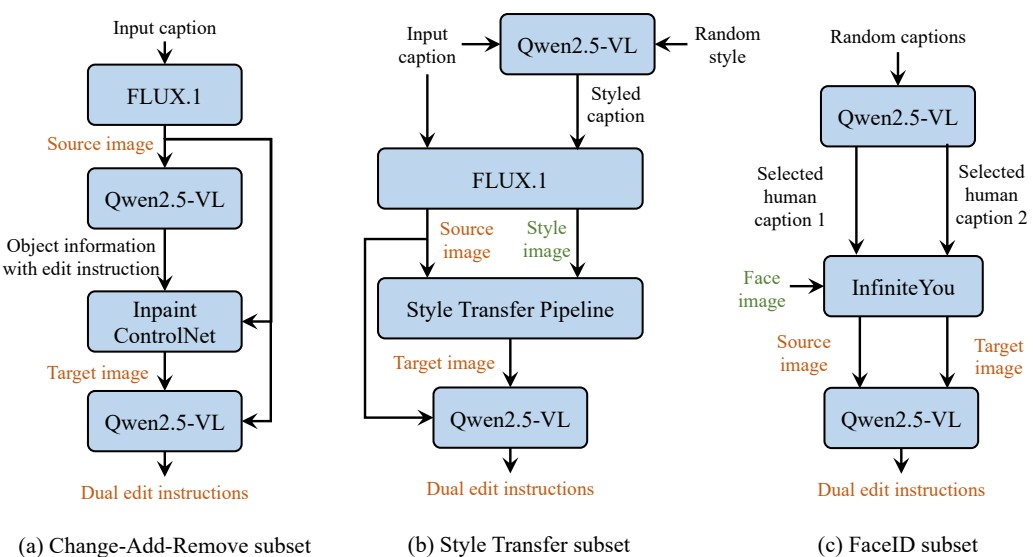

(a) Change-Add-Remove subset      (b) Style Transfer subset      (c) FaceID subset

Figure 7: The dataset construction pipeline for three subsets of ImagePulse.

**Bilingual Annotations**   Apart from the Chinese samples present in image understanding datasets, all aforementioned datasets are annotated exclusively in English. To endow Nexus-Gen with bilingual (Chinese-English) capabilities for both image generation and editing, additional Chinese image generation and editing samples are incorporated into the dataset. To this end, we perform Chinese re-annotation on several generation subsets (namely EliGen, FLUX-Aes, and FLUX-T2I) as well as the ImagePulse editing dataset. This process yielded a corpus of 2.5 million Chinese training samples, comprising the FLUX-ZH and ImagePulse-ZH subsets illustrated in Figure 6.

**Licencing**   To regulate the use of the constituent datasets, this work adheres to their respective licensing terms. The data distributions are covered by several open-source and restrictive licenses as follows: Apache 2.0 (covering Cambrian, EliGen, FLUX-Aes, blip3o-60k, Anyword, StyleBooth, and ImagePulse), MIT (FLUX-T2I and OmniEdit), and CC-BY-NC 4.0 (Laion, HQ-Edit, and UltraEdit). Furthermore, the Journey dataset is governed by a custom license that explicitly permits use only for non-commercial research.

## A.2   CONSTRUCTION PIPELINE FOR IMAGEPULSE

Each sample in ImagePulse contains: (1) pristine image pairs synthesized by FLUX.1-Dev Labs (2024), and (2) semantically-rich editing instruction generated by Qwen2.5-VL-72B Bai et al. (2025). For diverse editing tasks, the dataset is partitioned into three specialized subsets: Change-Add-Remove, Style Transfer, and FaceID. The workflow for each subset is illustrated in Figure 7.

**Change-Add-Remove**    This subset focuses on object-level image manipulations, including modifying object attributes (such as shape, material, and color), adding objects, and removing objects. The dataset construction pipeline is illustrated in Figure 7(a). First, we randomly sample a caption from the DiffusionDB Wang et al. (2022) dataset and synthesize a source image using the FLUX.1-Dev model. Next, the Qwen2.5-VL model extracts object information, comprising semantic descriptions and spatial locations, from this image and generates corresponding editing instructions. Subsequently, we use the extracted spatial locations and editing instructions to modify specified regions via Inpaint ControlNet Creative (2024), producing the target image. Finally, to maximize data utility, Qwen2.5-VL generates bidirectional editing instructions between the source and target images.

**Style Transfer**    This subset tackles the problem of image style transfer, with its construction process illustrated in Figure 7(b). First, a randomly selected input caption and a target style prompt are processed by Qwen2.5-VL to generate a style-transformed caption. Subsequently, the original input caption and the generated style-transformed caption are leveraged to synthesize the source image and the style image, respectively. Crucially, the style image exhibits significant structural divergence from the source image, rendering it unsuitable as the target image. To derive the target image, we develop a Style Transfer Pipeline integrating ControlNet, SDXL, InstantStyle, and IP-Adapter models. This pipeline effectively fuses the structural framework of the source image with the stylistic properties of the style image. Finally, we generate corresponding dual editing instructions.

**FaceID**    The unified architecture of Nexus-Gen facilitates tackling particularly challenging conceptual editing tasks by harnessing advanced image generation capabilities. To validate this capability, we construct the FaceID subset, which focuses on executing high-variation conceptual edits while preserving subject identity. The dataset construction process is illustrated in Figure 7(c). Initially, Qwen2.5-VL selects two captions containing human descriptions from a pool of randomly sampled image captions. Subsequently, a facial image is randomly sampled from the CelebA-HQ-Face Liu et al. (2015) dataset. Using these two captions and the facial image, we synthesize the source and target images via the InfiniteYou Jiang et al. (2025) model. These images exhibit identical subject identity while differing significantly in pose and scene context. Finally, dual editing instructions corresponding to the source and target images are generated by Qwen2.5-VL.

## B    IMPLEMENTATION DETAILS

Nexus-Gen employs Qwen2.5-VL Bai et al. (2025) as its autoregressive model and FLUX.1-Dev Labs (2024) as the generation and editing decoder, utilizing a multi-stage training strategy to optimize each component separately. The autoregressive model undergoes training within the ms-swift Zhao et al. (2024b) framework, comprising pretraining followed by an aesthetic fine-tuning stage. The generation and editing decoders are trained using Diffsynth-Studio ModelScope (2025). Detailed training hyperparameters are provided in Table 4. All models are trained using Nvidia A100 GPU. During inference, the generation decoder utilizes classifier-free guidance with a scale of 3.0.

## C    MORE QUANTITATIVE AND QUALITATIVE RESULTS

In this section, we present additional quantitative and qualitative results for Nexus-Gen, focusing on image generation and editing tasks.

### C.1    IMAGE EDITING PERFORMANCE ON MAGICBRUSH BENCHMARK

To validate the generalization editing capability of our model on unseen datasets, we conducted an additional evaluation on the MagicBrush benchmark Zhang et al. (2023). The evaluation results are presented in Table 5.

As shown in Table 5, Nexus-Gen achieves the highest CLIP-T score of 0.3107, which measures semantic accuracy and instruction-following ability, significantly outperforming all compared methods. On the L1 metric, which measures pixel-level similarity, our score is lower than that of the MagicBrush model, which was specifically trained on the MagicBrush dataset distribution.

Table 5: Editing performance comparison on the MagicBrush benchmark.

| Method | L1 ↓ | CLIP-T ↑ | CLIP-I ↑ | DINO ↑ |
|---|---|---|---|---|
| Open-Edit Liu et al. (2020) | 0.1430 | 0.2610 | 0.8381 | 0.7632 |
| HIVE Zhang et al. (2024) | 0.1092 | 0.2752 | 0.8519 | 0.7500 |
| InstructPix2Pix Brooks et al. (2023) | 0.1122 | 0.2764 | 0.8524 | 0.7428 |
| MagicBrush Zhang et al. (2023) | **0.0658** | 0.2812 | **0.9189** | **0.8655** |
| Nexus-Gen (Ours) | 0.1292 | **0.3107** | 0.8934 | 0.8268 |

## C.2 MORE QUALITATIVE RESULTS ON IMAGE GENERATION

Figure 8 displays representative high-fidelity images synthesized by Nexus-Gen, demonstrating the model's capability to accurately interpret semantic information from textual descriptions and translate them into visually coherent outputs. Owing to the incorporation of bilingual image generation datasets, Nexus-Gen is capable of processing inputs and generating outputs in both English and Chinese.

## C.3 MORE QUALITATIVE RESULTS ON IMAGE EDITING

The image editing capabilities of Nexus-Gen are demonstrated in Figure 9, which exhibits seamless handling of diverse workflows including subject addition, removal, replacement, color alteration, and style transfer. It can be observed that Nexus-Gen demonstrates remarkable proficiency in both preserving non-edited regions and executing editing instructions.

# D LIMITATIONS

Despite its capabilities in image understanding, generation, and editing, Nexus-Gen exhibits distinct limitations. First, with a total training dataset of 26 million samples, its scale remains substantially smaller than either specialized single-task models or unified counterparts trained on hyper-scale datasets. Consequently, the model may exhibit sensitivity to image generation prompts and often requires specific instruction templates for optimal editing performance. Second, unlike VAE latent spaces that enable precise pixel-level reconstruction, Nexus-Gen's unified image space operates primarily at the semantic feature level, resulting in inherent reconstruction fidelity limitations. Third, the visual reasoning capabilities of Nexus-Gen remains unexplored.

While the latter two limitations will be addressed in subsequent work, the issue of prompt robustness can be mitigated. Leveraging Nexus-Gen's comprehensive language reasoning and generation capabilities, we propose a Self-Prompt-Refinement strategy. This approach enables the model to autonomously rewrite its input prompts, thereby enhancing robustness to diverse prompt phrasing, as illustrated in Figure 10.

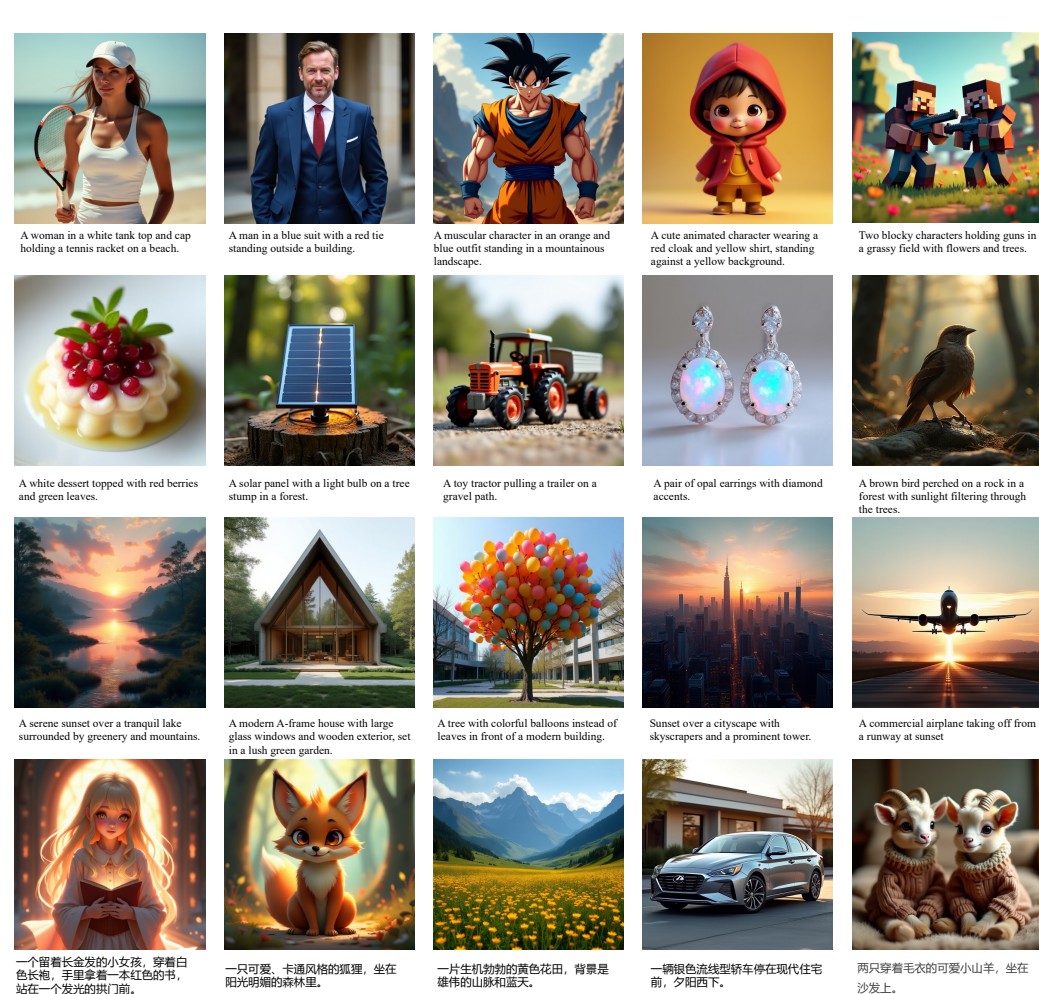

Figure 8: Qualitative image generation results of Nexus-Gen.

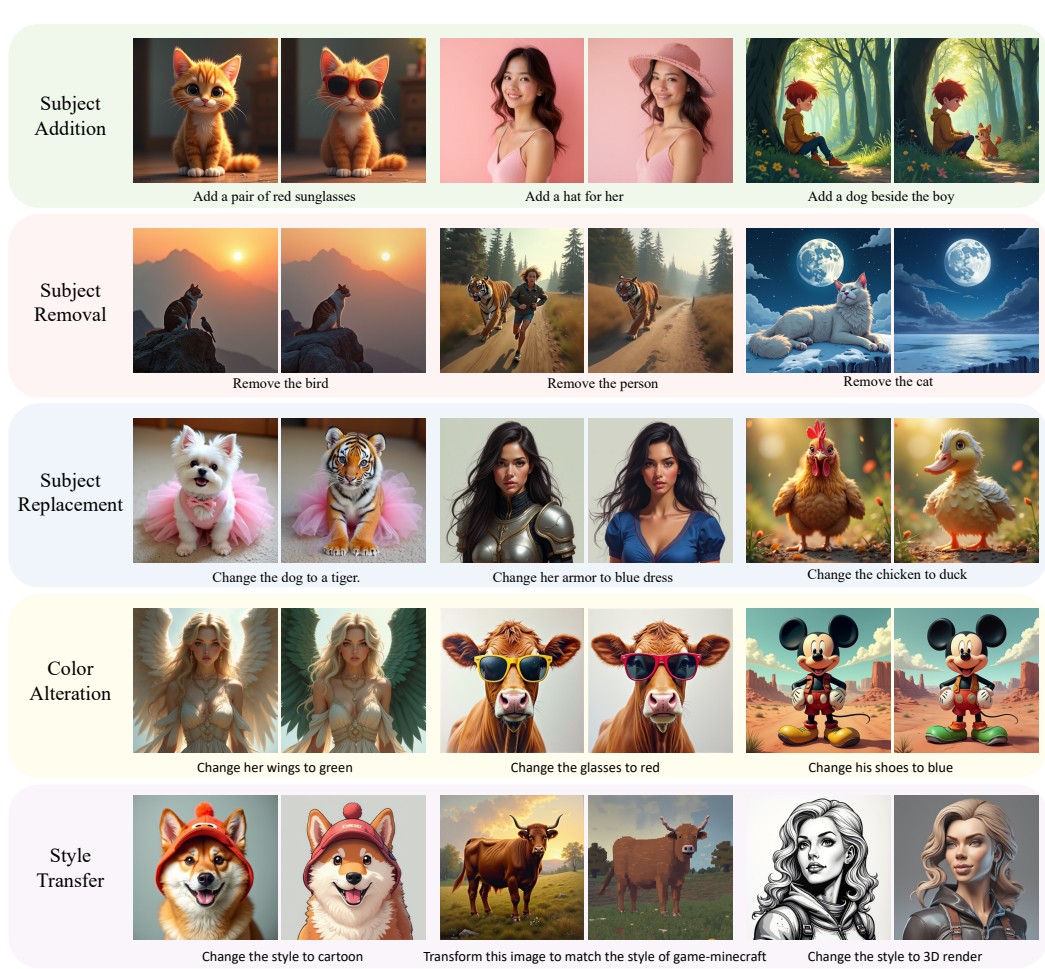

Figure 9: Qualitative image editing results of Nexus-Gen.

| Image Generation with Self-Prompt-Refinement | | Image Editing with Self-Prompt-Refinement | |
|---|---|---|---|
| User input | Generate an image according to the following description: A cat sits on a bench. | User input | Do image editing: A cat wearing sunglasses is lying on a bench |
| Step1: prompt refinement | Polish the image generation prompt and the polished result is: 

 A charming, fluffy cat sits gracefully on a wooden bench. The cat's fur appears soft and ...... In the background, there is a blurred natural setting......, capturing a quiet moment of solitude in nature. | Step1: prompt refinement | Given the image and a input instruction 'A cat wearing sunglasses is lying on a bench.', turn the prompt to a standard image editing prompt of the image. Example edit prompt format is 'Let someone smile.' 

 The reformatted standard prompt is : 
 Let the cat wear sunglasses and lie on a bench. |
| Step2: generation | Generate an image according to the extended prompt | Step2: editing | Do image editing: 

 Let the cat wear sunglasses and lie on a bench. |

Figure 10: Improving prompt robustness with self-prompt-refinement strategy.

