# OpenReview forum: "Nexus-Gen: Unified Image Understanding, Generation, and Editing via Prefilled Autoregression in Shared Embedding Space"
_ICLR.cc/2026/Conference — Submitted to ICLR 2026_

### Official Review · Reviewer_gxb3 · 2025-10-23

**Soundness:** 3
**Presentation:** 2
**Contribution:** 2
**Rating:** 2
**Confidence:** 4

**Summary:**

This paper presents Nexus-Gen, a unified multimodal framework that combines an autoregressive model (based on Qwen2.5-VL) with a diffusion-based vision decoder (FLUX). Nexus-Gen uses a shared continuous image embedding space as the interface between language and vision, enabling image understanding, text-to-image generation, and image editing within one model. To address the error accumulation problem in autoregressive prediction of continuous embeddings, this work proposes a prefilled autoregression mechanism, where learnable embeddings are used to keep training and inference consistent. The model is trained in three stages on a 26M multi-task dataset and achieves strong results across several benchmarks.

**Strengths:**

The paper introduces Nexus-Gen, a model designed to unify image understanding, generation, and editing within a single architecture. To address error accumulation during autoregressive prediction, the paper proposes a prefilled autoregression strategy, which is both simple and effective. The experimental results are comprehensive, covering understanding, generation, and editing benchmarks, with consistent improvements over comparable unified models. The work holds significant engineering and practical value, demonstrating a realistic pipeline for large-scale multimodal training.

**Weaknesses:**

- The technical novelty is somewhat limited. The unified-embedding approach and the combination of AR and diffusion decoders have been widely explored in recent works. The prefilled autoregression strategy can be viewed as a practical adaptation of teacher-forcing or prefix-tuning strategies rather than a fundamentally new modeling concept.

- While the prefilled strategy empirically reduces autoregressive MSE, the paper presents few variants (e.g., initialization schemes, shared vs. task-specific embeddings, or scheduled alternatives) or theoretical justification for when this heuristic may fail. More ablations would help establish robustness and applicability.

- The three-stage training pipeline requires a systematic study of stage ordering, joint vs. staged optimization, and data-mixing strategies to justify the optimality of this schedule. Without such analysis, the superiority of the specific design remains unclear.

- The model learns text-to-embedding during multitask pretraining and embedding-to-pixel during decoder adaptation. However, the paper does not clearly analyze how much of the generation quality comes from the autoregressive embedding predictor versus the diffusion decoder adaptation. Explicit ablations of the T2I objective would help clarify the contribution of each component.

- The paper employs large, mixed datasets and a multi-stage training strategy, yet it omits detailed resource accounting (GPU type/count and total GPU hours) and a precise breakdown of data licensing and composition. This makes it difficult for others to reproduce the work or make fair cost-benefit comparisons.

**Questions:**

Please see weaknesses.

---

> ### Author Response · Authors · 2025-11-17
> **Response to Reviewer gxb3 (Part 1/2)**
>
> Thank you for your valuable feedback regarding our paper's innovation and experiments. We will address the weaknesses you raised in logical order.
>
> ### **W1: About the novelty of this work.**
> > The technical novelty is somewhat limited. The unified-embedding approach and the combination of AR and diffusion decoders have been widely explored in recent works. The prefilled autoregression strategy can be viewed as a practical adaptation of teacher-forcing or prefix-tuning strategies rather than a fundamentally new modeling concept.
>
> Our main contributions and novelty are threefold, extending beyond the prefilled autoregression strategy you mentioned:
>
> 1.  **Error Accumulation & Prefilled Autoregression:** This strategy is one of our core novelties. We are the first work to explicitly identify the **error accumulation problem in unified models (when predicting continuous embeddings) and provide a viable solution**. Although inspired by teacher-forcing, we adapted it with **learnable tokens + positional encoding**, which has proven to be highly effective.
>
> 2.  **Unification of Three Tasks & Optimized Editing:** We unify three tasks—understanding, generation, and editing—with joint training, which is uncommon in prior works. Furthermore, to enhance detail preservation in the image editing task, we designed a novel dual-stream conditioning framework (shown in Figure 1(d)) that supplements the model with fine-grained information from the source image.
>
> 3.  **A Large-Scale Open-Source Dataset:** We constructed and release a **26.3 million-sample dataset**. All data from public sources were re-annotated for quality, and we additionally generated the entirely new, high-quality **ImagePulse dataset** for image editing. Given the scarcity of high-quality, open-source datasets for unified models, this represents a significant contribution to the field.
>
> ### **W5: Resource accounting and precise breakdown of data licensing and composition.**
> > The paper employs large, mixed datasets and a multi-stage training strategy, yet it omits detailed resource accounting (GPU type/count and total GPU hours) and a precise breakdown of data licensing and composition. This makes it difficult for others to reproduce the work or make fair cost-benefit comparisons.
>
> Thank you for your valuable suggestions. We supplement this information as follows and also in the Appendix:
>
> * **Resource Accounting:** We used **NVIDIA A100** GPUs. The Autoregressive Model was trained on a **128-GPU** cluster, with Large Scale Pretraining taking **14 days** and Aesthetic Fine-Tuning taking **4 days**. The Generation and Editing Decoders were trained on **8-GPU** nodes, taking **7 days** and **4 days**, respectively. Other implementation details are in Table 4 of the paper. We have added this information to the manuscript.
>
> * **Data Licensing:** The composition of our dataset is already listed in Appendix Figure 6. As per your suggestion, we have added the corresponding licenses:
>     * **Apache 2.0:** Cambrian, EliGen, flux_aes, blip3o-60k, Anyword, StyleBooth, and ImagePulse.
>     * **MIT:** flux_t2i and OmniEdit.
>     * **CC-BY-NC 4.0:** Laion, HQ-Edit, and UltraEdit.
>     * **Custom (Non-commercial Research):** Journey.

---

> ### Author Response · Authors · 2025-11-17
> **Response to Reviewer gxb3 (Part 2/2)**
>
> ### **W4: How much of the generation quality comes from the autoregressive embedding predictor versus the diffusion decoder adaptation?**
> > The model learns text-to-embedding during multitask pretraining and embedding-to-pixel during decoder adaptation. However, the paper does not clearly analyze how much of the generation quality comes from the autoregressive embedding predictor versus the diffusion decoder adaptation. Explicit ablations of the T2I objective would help clarify the contribution of each component.
>
> This is a crucial question. We have added the following comparative experiment on the GenEval benchmark to address it.
>
> * **Group 1:** The complete **AR + Diffusion Decoder**, representing the combined effect of our autoregressive embedding predictor and the adapted diffusion decoder.
> * **Group 2:** The **Single Diffusion Decoder** (our adapted decoder) used for image generation, but receiving image embeddings from images of **GPT-4o**.
> * **Group 3:** The **GPT-4o** evaluation results, serving as a reference.
>
> The results are as follows:
> | Method | Single | Two | Counting | Colors | Position | Color Attr. | Overall |
> | :--- | :--- | :--- | :--- | :--- | :--- | :--- | :--- |
> | GPT-4o (Group 3) | **0.99** | 0.92 | **0.85** | **0.92** | 0.75 | 0.61 | **0.84** |
> | Single Diffusion Decoder (Group 2) | 0.99 |**0.94** | 0.81 | 0.86 | 0.70 | **0.66** | 0.83 |
> | AR + Diffusion Decoder (Group 1) | 0.97 | 0.93 | 0.64 | 0.88 | **0.83** | 0.62 | 0.81 |
>
> As shown, the Single Diffusion Decoder (Group 2) scores very closely to GPT-4o (Group 3). **This demonstrates that our adapted decoder has minimal quality loss and performs the reconstruction task very well.**
>
> However, AR + Diffusion Decoder (Group 1) shows a noticeable performance difference compared to the other two groups. This indicates that the **semantic accuracy of the generated image is still more significantly influenced by the AR model's embedding prediction** rather than the diffusion decoder.
>
> ### **W3 & W4: Ablation Studies on prefilled strategy and three-stage training pipeline**
> > While the prefilled strategy empirically reduces autoregressive MSE, the paper presents few variants (e.g., initialization schemes, shared vs. task-specific embeddings, or scheduled alternatives) or theoretical justification for when this heuristic may fail. More ablations would help establish robustness and applicability.
> > The three-stage training pipeline requires a systematic study of stage ordering, joint vs. staged optimization, and data-mixing strategies to justify the optimality of this schedule. Without such analysis, the superiority of the specific design remains unclear.
>
> These are excellent and professional questions. However, given the large scale of our data (26.3M) and model, a full training run requires **2-3 weeks on 128 GPUs**. Consequently, a global search for detailed strategies, hyper-parameters, and data-mixing optimizations is extremely resource- and time-intensive, and could not be completed in the short term. We will consider adding this research in future work.
>
> Furthermore, for the training pipeline, current design follows a "AR-then-Diffusion" strategy, as Diffusion decoder is dependent on AR model. Referencing other large model training paradigms:
> 1.  The AR model is first pretrained on all data, then fine-tuned for the quality of the image distribution.
> 2.  The Diffusion decoders are trained separately by task. Since the Generation and editing decoders do not share parameters, their training order is not strictly sequential.
>
> **We have made substantial revisions to address your concerns. As such, we kindly hope that you will consider increasing our overall score.**

---

### Official Review · Reviewer_fhHJ · 2025-10-28

**Soundness:** 3
**Presentation:** 3
**Contribution:** 2
**Rating:** 4
**Confidence:** 4

**Summary:**

This paper presents Nexus-Gen, a unified multimodal generative framework that integrates image understanding, generation, and editing within a shared image embedding space. The model bridges autoregressive and diffusion paradigms to leverage their complementary strengths. To address error accumulation in autoregressive embedding prediction, the authors propose a prefilled autoregression strategy that aligns training and inference by initializing image tokens with learnable embeddings. Trained on a large-scale dataset of 26.3M samples, Nexus-Gen achieves state-of-the-art performance across multimodal benchmarks.

**Strengths:**

1.The paper proposes a comprehensive unified framework that integrates image understanding, generation, and editing, effectively bridging autoregressive and diffusion modeling paradigms within a shared embedding space.

2.The prefilled autoregression strategy is a well-motivated solution to mitigate error accumulation and exposure bias in autoregressive prediction, improving training–inference consistency.

3.The paper demonstrates strong empirical performance on multiple multimodal benchmarks, supported by large-scale experiments (26.3M samples) and clear ablation analyses.

**Weaknesses:**

1.The architectural novelty appears limited — combining autoregressive and diffusion components within a shared embedding space has been explored in several prior unified or hybrid generative models. The framework design is well-engineered but does not represent a fundamentally new modeling paradigm.

2.The proposed prefilled autoregression technique seems only loosely related to the unified multimodal objective. It primarily addresses the exposure bias issue in autoregressive prediction, which could, in principle, also apply to standard language models. Its contribution to multimodal unification therefore appears indirect.

3.The paper lacks clarity on training cost and computational efficiency. Given the scale of 26.3M training samples, it is important to report details such as GPU hours, batch size, or hardware configuration to assess the practical feasibility of reproducing or extending the work.

**Questions:**

see the weakness.

---

> ### Author Response · Authors · 2025-11-17
> **Response to Reviewer fhHJ**
>
> Thank you very much for your insightful review. We have addressed the Weaknesses you pointed out one by one as follows:
>
> ### **W1 & W2: Novelty and Contribution of This Work**
>
> > The architectural novelty appears limited — combining autoregressive and diffusion components within a shared embedding space has been explored in several prior unified or hybrid generative models. The framework design is well-engineered but does not represent a fundamentally new modeling paradigm.
> > The proposed prefilled autoregression technique seems only loosely related to the unified multimodal objective. It primarily addresses the exposure bias issue in autoregressive prediction, which could, in principle, also apply to standard language models. Its contribution to multimodal unification therefore appears indirect.
>
>
>
> Our work focuses on unified architectures combining AR and diffusion models, which is indeed a popular paradigm. Though we are not the first to propose such an architecture, we have made distinct and significant contributions in at least three areas:
>
> 1. **Error Accumulation & Prefilled Autoregression:** We are the first to identify and solve the **error accumulation problem specific to predicting continuous embeddings** in unified models. This is fundamentally different from standard exposure bias. The error accumulation we address stems from the unavoidable MSE error in continuous predictions, whereas exposure bias originates from the misclassification of discrete tokens. This difference in origin makes the solutions non-interchangeable. **Therefore, the contribution of Prefilled Autoregression to multimodal unification is direct, as it tackles a problem traditional exposure bias methods are not suited for.**
>
> 2. **Unification of Three Tasks and Optimized Image Editing:** We unify three tasks—understanding, generation, and editing—with joint training, which is uncommon in prior works. Furthermore, to enhance detail preservation in the image editing task, we designed a novel dual-stream conditioning framework (shown in Figure 1(d)) that supplements the model with fine-grained information from the source image.
>
> 3. **A Large-Scale Open-Source Dataset:** We constructed and will release a 26.3 million-sample dataset. All data are re-annotated for quality, and we additionally generated the entirely new, high-quality ImagePulse dataset for image editing. Given the scarcity of high-quality, open-source datasets for unified models, this represents a significant contribution to the field.
>
> ### **W3: Clarity on Training Cost and Computational Efficiency**
>
> > The paper lacks clarity on training cost and computational efficiency. Given the scale of 26.3M training samples, it is important to report details such as GPU hours, batch size, or hardware configuration to assess the practical feasibility of reproducing or extending the work.
>
> Thank you for pointing this out; this is a very important supplement.
>
> We have indeed listed the training settings for each stage in **Table 4 in the Appendix**, including Learning Rate, Batch Size , Total Steps, etc.
>
> For the hardware resources and training times, we supplement here and in the Appendix:
> * **Hardware:** We used **NVIDIA A100** GPUs for all training.
> * **Autoregressive Model Training:** Conducted on a **128-GPU** cluster.
>     * Large Scale Pretraining: **14 days**.
>     * Aesthetic Fine-Tuning: **4 days**.
> * **Decoder Training:** Both decoders were trained on a **8-GPU** node.
>     * Generation Decoder: **7 days**.
>     * Editing Decoder: **4 days**.
>
> Other implementation details can be found in Table 4.
>
> **We have made substantial revisions to address your concerns. As such, we kindly hope that you will consider increasing our overall score.**

---

### Official Review · Reviewer_muUH · 2025-10-28

**Soundness:** 4
**Presentation:** 4
**Contribution:** 2
**Rating:** 6
**Confidence:** 4

**Summary:**

This paper presents Nexus-Gen, a novel unified model for image understanding, generation, and editing. The authors identify two key limitations in prior work (e.g., AR+VAE models, AR+Diffusion models): 1) poor image synthesis quality and 2) error accumulation during the autoregressive (AR) prediction of continuous image embeddings used to condition diffusion decoders. Nexus-Gen addresses this by proposing a unified architecture that bridges a powerful AR model (Qwen2.5-VL) and a diffusion decoder (FLUX.1-Dev) through a shared continuous image embedding space.

**Strengths:**

1.  The paper's core technical contribution is a novel and highly effective method for mitigating error accumulation in AR models predicting continuous embeddings. This is a key problem in AR+Diffusion hybrids, and the solution is well-supported by a strong ablation (Fig 5).
2.  The architecture is clean and effective. The use of a shared embedding space is standard, but the decision to employ two *separate*, specialized decoders (one for generation, one for editing) is a smart design choice that is well-justified by ablations (Fig 4) and leads to high-fidelity editing.
3. The curation of the 26.3M dataset, and particularly the new `ImagePulse` editing dataset, is a significant resource contribution that addresses a known lack of high-quality editing data.

**Weaknesses:**

1. The image editing performance (Table 3) is evaluated on a test set from their own `ImagePulse` dataset. While this dataset is a contribution, the pipeline used to create it (FLUX.1, Qwen-VL, ControlNet) could introduce biases that their model is well-suited to, but which may not generalize. It would be more convincing to also show evaluations on a hold-out from an existing, standard benchmark (e.g., Magic Brush, or a set from UltraEdit not used in training).
2. The authors honestly state this as a limitation. While the model excels on "understanding" benchmarks (VQA, MMMU), these are largely recognition- and knowledge-based. The paper does not explore more complex, emergent capabilities like in-context learning with interleaved images/text or step-by-step visual reasoning, which are key motivators for unified models.
3. The model's strong performance is built on top of very powerful, state-of-the-art base models (Qwen2.5-VL-7B and FLUX.1-Dev). This makes it slightly difficult to disentangle how much of the final performance comes from the "glue" (the novel contributions) versus the power of the components. This is a common issue in SOTA research but worth noting.

**Questions:**

1.  Could you clarify the nature of the "learnable embeddings" used in the prefilled autoregression strategy? Are they static (like positional encodings), or are they dynamically modulated by the text prompt? How are they initialized and trained?
2.  To address the potential for "overfitting" to your own data-generation pipeline, could you provide editing evaluation results on an existing, external benchmark (e.g., Magic Brush) that was not part of your training mix?
3.  The architecture seems to naturally support in-context learning and interleaved image-text inputs. Have you performed any preliminary experiments on this? How does the model perform on such tasks compared to models like Emu3?
4.  What is the computational overhead (e.g., in FLOPS or inference time) of the prefilled autoregression strategy compared to the naive token-by-token AR baseline?

---

> ### Author Response · Authors · 2025-11-17
> **Response to Reviewer muUH (Part 1/2)**
>
> Thank you for your patient review and insightful suggestions. We have addressed the weaknesses and questions you raised one by one:
>
> ### **Q1: Details about the prefilled autoregression strategy.**
> > "Could you clarify the nature of the "learnable embeddings" used in the prefilled autoregression strategy? Are they static (like positional encodings), or are they dynamically modulated by the text prompt? How are they initialized and trained?"
>
> **Reply:** Thank you for the question. These "learnable embeddings" are **81 learnable token embeddings (like tokens in a vocabulary)** of the model, with a dimension of $81 \times 3584$. They are **initialized randomly**. During both training and inference, they are "prefilled" into the sequence and **utilize the same 9x9 grid relative position encoding as the image understanding task** to acquire spatial information.
>
> ### **Q2 & W1: Evaluations on a hold-out MagicBrush benchmark.**
> > "To address the potential for "overfitting" to your own data-generation pipeline, could you provide editing evaluation results on an existing, external benchmark (e.g., Magic Brush) that was not part of your training mix?"
>
> **Reply:** Your insight regarding data bias is critical. As you suggested, we evaluated Nexus-Gen on the hold-out test set of **MagicBrush** and compared it with other methods. The results are as follows:
>
> | Method | L1 $\downarrow$ | CLIP-T $\uparrow$ | CLIP-I $\uparrow$ | DINO $\uparrow$ |
> | :--- | :---: | :---: | :---: | :---: |
> | Open-Edit | 0.1430 | 0.2610 | 0.8381 | 0.7632 |
> | HIVE | 0.1092 | 0.2752 | 0.8519 | 0.7500 |
> | InstructPix2Pix | 0.1122 | 0.2764 | 0.8524 | 0.7428 |
> | MagicBrush | **0.0658** | 0.2812 | **0.9189** | **0.8655** |
> | **Nexus-Gen (Ours)** | 0.1292 | **0.3107** | 0.8934 | 0.8268 |
>
> As you can see, **Nexus-Gen achieves the highest score on the CLIP-T metric, which measures semantic accuracy (i.e., instruction-following ability)**.
>
> As our model was not trained on the MagicBrush data distribution, its scores on metrics evaluating pixel-level similarity (like L1) are lower than the MagicBrush model, which was trained on that specific distribution. This is expected. We have added this result to the revised version of our paper (Appendix, Section C.1).
>
> ### **Q3 & W2: More complex, emergent capabilities of unified models.**
> > "While the model excels on "understanding" benchmarks (VQA, MMMU)... The paper does not explore more complex, emergent capabilities like in-context learning with interleaved images/text or step-by-step visual reasoning, which are key motivators for unified models."
>
> **Reply:** Your understanding of unified models is very thorough. The ultimate goal of unified models is indeed the mutual promotion of understanding and generation capabilities. We conducted preliminary explorations on this:
>
> 1.  **From "Understanding promotes Generation", the effect is significant.** Nexus-Gen significantly improves the generation performance.
>     * **First, on GenEval benchmark,** the original FLUX.1-Dev scores only 0.66, whereas Nexus-Gen achieves **0.81**. This proves the AR framework's semantic understanding directly boosts the generator's semantic accuracy
>     * **Furthermore,** as shown in Appendix D, we also explored **leveraging the model's own reasoning capabilities to refine image generation** (e.g., via Prompt Refinement).
> 2.  **From "Generation promotes Understanding", the effect is not yet evident, which is currently the common phenomenon of this area.** We observed that introducing generation/editing tasks had a slight negative impact on the understanding benchmark scores (Table 1). We also attempted experiments on step-by-step reasoning by generating images, but we have not yet observed clear gains, as these tasks demand extremely high spatial precision from the generated images. However, **this is a problem faced by the entire field collectively and a key direction for future breakthroughs.**

---

> ### Author Response · Authors · 2025-11-17
> **Response to Reviewer muUH (Part 2/2)**
>
> ### **Q4: Computational overhead of prefilled autoregression over token-by-token AR.**
> > "What is the computational overhead (e.g., in FLOPS or inference time) of the prefilled autoregression strategy compared to the naive token-by-token AR baseline?"
>
> **Reply:** Thank you for your attention to the computational details. Assuming $N_0$ prefix tokens, $N_1$ prefilled tokens.
>
> * **Practical Inference Latency:** **The Prefilled strategy is much faster in practice**. Naive AR must be executed serially $N_1$ times (to decode $N_1$ tokens), whereas our Prefilled strategy **compute all $N_1$ tokens in a single parallel forward pass**.
> * **Theoretical FLOPs:** The Prefilled strategy does have higher theoretical FLOPs. In naive AR (serial), the $i$-th token only attends to $N_0+i-1$ tokens. In our Prefilled strategy, *each* of the $N_1$ tokens attends to all $N_0+N_1$ tokens.
>
> Assuming the FLOPs of one Query attending to $k$ Keys/Values is $k \times F_0$, the theoretical flops is:
>
> * **Naive AR (Causal):** $F_{AR} \approx \sum_{i=1}^{N_1} (N_0 + i - 1) \times F_0 \approx F_0 \times (N_0 \cdot N_1 + N_1^2 / 2)$
> * **Prefilled AR (Parallel):** $F_{Prefill} = \sum_{i=1}^{N_1} (N_0 + N_1) \times F_0 = F_0 \times (N_0 \cdot N_1 + N_1^2)$
>
> ### **W3: Disentangle how much of the final performance comes from the "glue" (the novel contributions) versus the power of the components.**
> > "The model's strong performance is built on top of very powerful, state-of-the-art base models (Qwen2.5-VL-7B and FLUX.1-Dev). This makes it slightly difficult to disentangle how much of the final performance comes from the "glue" (the novel contributions) versus the power of the components."
>
> **Reply:** This is a very valuable and pertinent point. We address this in two parts:
>
> 1. **Current Experimental Evidence:** The strongest evidence for disentangling the value of Nexus-Gen comes from the **GenEval benchmark**. The original FLUX.1-Dev scores only 0.66. **This significant semantic improvement (0.81 vs 0.66)** clearly demonstrates the improved generation ability.
>
> 2. **Future Work:** We fully agree on the importance of further disentangling the architectural contribution. Therefore, we will advance this in our subsequent work, eg. swap out components and compare the performance changes.
>
> **We have made substantial revisions to address your concerns. As such, we kindly hope that you will consider increasing our overall score.**

---

### Official Review · Reviewer_5r2C · 2025-10-30

**Soundness:** 2
**Presentation:** 3
**Contribution:** 2
**Rating:** 4
**Confidence:** 3

**Summary:**

This paper proposes Nexus-Gen, a unified multi-modal large language model (MLLM). The core innovations lies in the Prefilled Autoregression Strategy. By initializing image tokens with learnable embeddings (instead of relying on real previous tokens during training), this strategy aligns training and inference processes. Experiments are conducted on MMMU, GenEval, and ImagePulse.

**Strengths:**

1. Originality: The prefilled autoregression strategy is a non-trivial improvement over naive autoregressive generation.​

2. Clarity: Technical components (e.g., dataset construction, training stages) are described with specific numbers (e.g., 430K aesthetic fine-tuning samples) and clear definitions (e.g., ImagePulse’s three subsets), making the work reproducible.​

**Weaknesses:**

1. Lack of experiments to validate the effectiveness of the core innovation, i.e. prefilled autoregression strategy. For example, please report the result w/o prefilled autoregression strategy in Table1-3.

2. Omitting comparisons with some recent work, e.g. OpenUni [a], BLIP3-o-8B [b], and Bagel [c]. BLIP3-o achieves 0.84, OpenUni 0.86 and Bagel 0.88 on GenEval. These scores are higher than that of the proposed method (0.81).

3. Prompt Robustness Limitation: The paper acknowledges that Nexus-Gen requires specific instruction templates (e.g., structured prompts for editing), but it does not quantify this limitation (e.g., error rates with unstructured prompts) or propose mitigation strategies. This reduces usability for real-world scenarios where users may input arbitrary prompts.​

[a] Wu S, Wu Z, Gong Z, et al. OpenUni: A Simple Baseline for Unified Multimodal Understanding and Generation[J]. arXiv preprint arXiv:2505.23661, 2025.
[b] Chen J, Xu Z, Pan X, et al. Blip3-o: A family of fully open unified multimodal models-architecture, training and dataset[J]. arXiv preprint arXiv:2505.09568, 2025.
[c] Deng C, Zhu D, Li K, et al. Emerging properties in unified multimodal pretraining[J]. arXiv preprint arXiv:2505.14683, 2025.

**Questions:**

See above.

---

> ### Author Response · Authors · 2025-11-17
>
> Thank you for your patient review and detailed suggestions. We have addressed the weaknesses you pointed out one by one as follows:
>
> ### **W1: Lack of experiments to validate the effectiveness of the prefilled autoregression strategy.**
> > Lack of experiments to validate the effectiveness of the core innovation, i.e. prefilled autoregression strategy. For example, please report the result w/o prefilled autoregression strategy in Table1-3.
>
> **Reply:** Thank you for acknowledging the strategy's originality. We have fundamentally validated its necessity via the **MSE loss comparison in Figure 5**.
>
> As shown, naive autoregression (blue line) suffers from severe error accumulation during inference (MSE 0.5543) compared to training (0.2574). Our strategy (red line) **significantly reduces the inference loss to 0.4300**.
>
> More importantly, **without the prefilled strategy, the model fails to converge or produce coherent images** due to this rapid error accumulation. Therefore, reporting benchmark results in Tables 1-3 for a "w/o prefilled" model is infeasible, as it is non-functional for these tasks.
>
> ### **W2: Comparisons with some recent works on GenEval.**
> > Omitting comparisons with some recent work, e.g. OpenUni [a], BLIP3-o-8B [b], and Bagel [c]. BLIP3-o achieves 0.84, OpenUni 0.86 and Bagel 0.88 on GenEval. These scores are higher than that of the proposed method (0.81).
>
> **Reply:** Thank you for mentioning these recent works.
>
> 1. We acknowledge these models achieve higher scores on GenEval. However, we believe a **single benchmark score should not be the sole metric** for evaluating a unified model's contribution. (For instance, the powerful GPT-Image also achieved only 0.84 on GenEval, lower than Bagel).
> 2.  **Nexus-Gen's unique contributions lie in:** (1) Identifying and solving the **error accumulation problem** in continuous embedding prediction; (2) Unifying understanding, generation, and ***editing*** (the models you mentioned focus mainly on understanding and generation); and (3) Contributing a **large-scale, 26.3M-sample open-source dataset** (especially the high-quality ImagePulse editing data), which is a significant asset for the community.
>
> ### **W3: Mitigation strategy for prompt robustness limitation.**
> > Prompt Robustness Limitation: The paper acknowledges that Nexus-Gen requires specific instruction templates (e.g., structured prompts for editing), but it does not quantify this limitation (e.g., error rates with unstructured prompts) or propose mitigation strategies. This reduces usability for real-world scenarios where users may input arbitrary prompts.​
>
> **Reply:** This is a very valuable suggestion. **We have added a mitigation strategy to the Appendix Section D**.
>
> We propose a **"Self-Prompt-Refinement" strategy (Figure 10 in the revised paper) that leverages the model's own language capabilities**: the model first rewrites a user's arbitrary prompt into an optimal structured instruction before execution. This improves usability and success rates while maintaining flexibility.
>
> **We have made substantial revisions to address your concerns. As such, we kindly hope that you will consider increasing our overall score.**

---

### Meta-Review · Area_Chair_d5zH · 2026-01-07

**Summary:**

The AC carefully read the paper and the full discussion. The submission received mixed reviews (initial scores: 4, 6, 4, 2). Reviewers generally agreed that the paper proposes a comprehensive unified framework integrating image understanding, generation, and editing, and that it helps bridge autoregressive and diffusion modeling paradigms within a shared embedding space. The prefilled autoregression strategy was viewed as a well-motivated approach to mitigate error accumulation and exposure bias in autoregressive prediction, improving training–inference consistency. The paper also reports strong empirical performance on multiple multimodal benchmarks, supported by large-scale experiments (26.3M samples) and clear ablation studies.

However, the main concerns center on limited novelty: the unified-embedding formulation and the combination of AR and diffusion decoders have been widely explored in recent work. The prefilled autoregression strategy can be seen as a practical adaptation of teacher forcing or prefix-tuning rather than a fundamentally new modeling concept. Reviewers also noted missing comparisons with some recent methods, and the paper does not provide strong evidence that generation quality is attributable to the autoregressive embedding predictor rather than the diffusion decoder adaptation. These issues raise questions about robustness, generality, and practical usability. As a result, it remains unclear whether the proposed method and pipeline deliver clear, meaningful benefits for current or future unified multimodal models. With the overall scores trending toward rejection and the core issues seemingly unlikely to be resolved through discussion, I am inclined to recommend rejection.

**Reviewer Concerns:**

Some concerns regarding prompt robustness (Reviewer 5r2C), as well as the risk of overfitting to the custom dataset and the need for external editing benchmarks such as MagicBrush (Reviewer muUH), have been addressed. However, several non-negligible issues remain:

Two reviewers (e.g., fhHJ and gxb3) noted limited novelty. In particular, combining autoregressive and diffusion components within a shared embedding space has been explored in multiple prior unified or hybrid generative models. While the framework is well engineered, it does not appear to introduce a fundamentally new modeling paradigm.

There are also concerns about the validation of the prefilled autoregression strategy raised by Reviewers 5r2C and gxb3. Relatedly, reviewers questioned whether the reported generation quality should be attributed to the autoregressive embedding predictor rather than the diffusion-decoder adaptation. Although the authors report an MSE loss, it does not strongly support this claim. The authors also argue that autoregression without prefilling cannot successfully learn image-generation embeddings, but this appears inconsistent with prior work showing that autoregressive modeling without prefilling can work in the embedding space (e.g., Fluid).

Overall, it remains unclear whether the proposed approach delivers clear or meaningful benefits for current or future unified models.

**Reviewer Scores:**

All reviewers are most likely to keep their scores unchanged, since the two main concerns above have not been adequately addressed.

---

### Decision · Program_Chairs · 2026-01-26

Reject